# Effect of Local Pressurization on Microstructure and Mechanical Properties of Aluminum Alloy Flywheel Housing with Complex Shape

**DOI:** 10.3390/ma17010106

**Published:** 2023-12-25

**Authors:** Qiang Chen, Ning Ge, Jufu Jiang, Minjie Huang, Mingxing Li, Ying Wang, Jian Dong, Changjie Ding, Dechao Zou

**Affiliations:** 1Southwest Technology and Engineering Research Institute, Chongqing 400039, China; 2009chenqiang@163.com; 2School of Materials Science and Engineering, Harbin Institute of Technology, Harbin 150001, China; 13735771019@163.com (N.G.); minjiehuang199607@163.com (M.H.); lmxdw123456@163.com (M.L.); 22b909131@stu.hit.edu.cn (J.D.); 3School of Mechatronics Engineering, Harbin Institute of Technology, Harbin 150001, China; 4Dalian Innovation Die-Casting Co., Ltd., Dalian 116600, China; dhmdcj@163.com (C.D.); zoudechao@innovation-dalian.com (D.Z.)

**Keywords:** squeeze casting, AlSi9Mg aluminum alloy, microstructures, mechanical properties, local pressurization, large wall thickness difference

## Abstract

In this work, squeeze casting experiments of flywheel housing components with a large wall thickness difference and a complex shape were carried out with AlSi9Mg aluminum alloy. The defects, microstructures, and mechanical properties under different process parameters were investigated. Furthermore, the local pressurization process was applied to the thick-walled positions to force-feed the cast defects. The mechanical properties and microstructures at these positions were analyzed. The results showed that the surface quality of formed components was good and that local pressurization could effectively reduce the shrinkage cavity and shrinkage porosity in thick walls, but the scope and effect of forced feeding were limited. The optimum process parameters were a pouring temperature of 650 °C, a specific pressure of 48 MPa, a mold temperature of 220 °C, a local pressurization of 800 MPa, and pressure delay times of 15 s (side A) and 17 s (side B). The ultimate tensile strength, yield strength, and elongation of the formed component under validation experiments of the optimum process parameters were 201 MPa, 103 MPa, and 5.1%. Meanwhile, the fine grains of primary α-Al were mainly rosette and equiaxed grains, and the average grain size was about 40 μm. The microstructure of the eutectic silicon was acicular and was prone to segregation under pressure. According to profile morphology, the positions after pressurization were divided into a deformation zone, a direct action zone, and an indirect action zone. The coexistence of as-cast and plastic deformation microstructures was observed. The effect of local pressurization mainly involved a change in the solidification process, plastic deformation, and forced feeding.

## 1. Introduction

The huge automotive industry has caused great energy consumption. It is hoped that upstream manufacturing can control energy consumption, and lightweight materials have become the consensus for the development of the automotive industry. Structural optimization and seeking lighter alternative materials are important methods of achieving this, and the method of using lightweight materials is the most direct and effective [1,2].

The overall structure of an automobile can be divided into the body, chassis, engine, and electrical equipment [2]. Among automotive parts, aluminum alloy parts are widely used. Aluminum alloy can be seen in the body, chassis, and engine of metal structures, such as the wheel hub, steering knuckle, front subframe, control arm, engine box, and other parts [3]. The most common manufacturing methods of these aluminum alloy parts are casting and forging, followed by machining. However, such methods not only have high production costs but also cause material waste and other problems [4]. In order to overcome the above problems, the near-net forming method of aluminum alloy parts has received more attention. Meanwhile, to improve the performance of aluminum alloy parts, new near-net forming methods have been applied to produce sound parts with complex structures [5].

Squeeze casting is a green, economical, and feasible near-net forming method for aluminum alloy parts that combines the characteristics of casting liquid filling and forging plastic deformation [6]. In squeeze casting, the mold filling, solidification crystallization, and feeding processes are completed under pressure to fill the cavity smoothly, refine grain size, improve microstructure, control casting defects, and, finally, obtain high-performance components, which can be further enhanced by subsequent heat treatment [7]. At present, squeeze casting is widely used in the near-net forming of automotive parts of different sizes and occupies a place in the parts market [8]. Although squeeze casting has many obvious advantages in producing light metal structural parts, its potential can be brought into full play only after optimizing the process. A series of experimental studies are needed for specific products to find the best forming method and optimize the process parameters. The final mechanical properties and microstructure of the obtained castings also depend on the process parameters. Pastirčák et al. [9] investigated the effect of the process parameters of squeeze casting on the microstructure and mechanical properties of AlSi12 and AlSi7Mg0.3 alloys. Dao et al. [10] and Wang et al. [11] studied the ZL104 automotive connecting rod by squeeze casting and investigated the effects of process parameters such as the extrusion force, mold preheating temperature, and casting temperature on the microstructure and mechanical properties of the connecting rod, respectively. Li et al.’s trials produced a large-scale Al-Cu-Mn alloy automotive wheel [12] and supporting frame part [13] according to the designed indirect squeeze casting manufacturing system and indirect squeeze casting process. Senthil and Amirthagadeswaran [14] investigated and optimized the squeeze casting process for manufacturing asymmetric AC2A aluminum alloy automotive wheel cylinders by the Taguchi method. Under the best conditions, the parts produced were free of defects and intact, and the defects were completely eliminated.

As a key component of automotive engine systems, flywheel housing with a large wall thickness difference and complex shape requires high comprehensive performance. In the production process, shrinkage porosity and shrinkage cavity defects often appear at the thick-walled positions, which directly affects the performance. Therefore, manufacturing technologies of aluminum alloy flywheel housing with low costs and high performance by effectively eliminating the defects at the thick-walled positions have attracted great attention. In this work, a novel technology called local pressurization was adopted. Chen et al. [15] used a local-loading squeeze casting process to form ZL104 aluminum alloy flywheel housing with a large wall thickness difference and a complex shape. The ultimate tensile strength and yield strength of the ZL204 aluminum alloy flywheel housing after T6 heat treatment reached 312 MPa and 251 MPa, respectively. During the heat treatment process, the morphology of Si was spheroidized to form fine Si particles, and the microstructure underwent incomplete dynamic recrystallization. However, there are few reports on its theoretical research and engineering applications. Local pressurization forming needs to be further investigated.

In this work, the process parameter variables affecting the quality, microstructures, and properties of squeeze casting flywheel housing were investigated, and the influence laws and the optimum process parameters were explored. Meanwhile, the mechanical properties and microstructures at the positions with local pressurization were emphatically analyzed. In addition, the effect of local pressurization was explored.

## 2. Experimental Procedures

### 2.1. Materials

The material used in this work was AlSi9Mg aluminum alloy, which is one of the most commonly used aluminum alloys in the automotive industry, mainly in engine parts. It has excellent casting performance and corrosion resistance. A ZSX Primus II X-ray (Kabuskiki kaisha, Nagoya, Japan) fluorescence spectrometer was used to determine the chemical composition of the alloy, and the chemical composition (wt. %) is shown in Table 1. The liquidus and solidus temperatures of the alloy were measured by differential scanning calorimetry (DSC) by a NETZSCH TG-DTA/DSC (NETZSCH, Selbu, Germany) integrated thermal analyzer. A cylindrical sample was inserted into the analyzer, heated to 700 °C at 10 °C/min in an argon gas atmosphere, and then cooled to room temperature at an identical speed. Figure 1 shows the DSC plot of the AlSi9Mg aluminum alloy. It reveals that the liquidus and the solidus temperatures of the alloy were 622.9 °C and 564.8 °C, respectively. For the integral of the surface surrounded by the peak, the value was 363.2 J/g, that is, 363.2 J of heat released per gram of melted alloy.

### 2.2. Methods

#### 2.2.1. Orthogonal Experimental Scheme

The orthogonal experiment is an efficient experimental method [16] that can replace the whole experiment with a partial experiment and then provide an understanding of the overall experiment by analyzing the partial experiment. Its core is the design of an orthogonal array, which has orthogonality, representativeness, and comprehensive comparability.

In this work, five factors, including pouring temperature, mold temperature, specific pressure, local specific pressure, and pressure delay time, were considered in the forming experiments, and each factor had three levels, as shown in Table 2. The orthogonal experimental scheme designed in this work is shown in Table 3. The *L*_18_(3^7^) orthogonal array was selected in the orthogonal experiment of five factors and three levels.

#### 2.2.2. Forming Process

The process of the forming experiment included the smelting and refining of the aluminum alloy, the quantitative pouring of the aluminum alloy melt, the squeeze casting of the components, and so on. The experimental process of the squeeze casting in this work is shown in Figure 2. Aluminum alloy was centrally smelted and refined in the tower smelting furnace. The alloy melt was transferred to the quantitative pouring furnace through the crucible, and then the alloy melt was poured into the pressure chamber through the runner of the quantitative pouring furnace. After the mold was closed, the alloy melt was squeezed into the mold cavity, and the pressure was maintained. After a certain delay, the oil cylinders designed at the mold drove the local pressurization devices to pressurize the thick-walled positions locally, and then the local pressurization devices were pulled out. Finally, the mold was opened, and the component was ejected from the mold.

The squeeze casting mold of the flywheel housing is shown in Figure 2 and had two parting surfaces, the pouring surface and the mold joint, respectively. The material of the mold core was steel H13. Three groups of local pressurization and core pulling devices were designed. The forming surface of the sliding block was complicated by bulges and deep cavities. The local pressurization adopted the method of multi-point with six rods at side A and side B and a single rod at side C, as shown in Figure 2. The rods for local pressurization arranged in corresponding positions moved forward relative to the sliding block under the action of the mold cylinders to complete the local pressurization.

#### 2.2.3. Characterizations

The surface quality and internal macroscopic morphology of the components were observed, and the mechanical properties of the components at different positions and directions were tested. Due to the large overall size of the flywheel housing component (i.e., 576 mm × 547 mm × 184 mm) and the high weight of about 20.4 kg, samples needed to be taken at different positions and dimensions to characterize the overall mechanical properties of the components. The tensile specimens were machined according to the national standard GB/T228.1-2010 [17]. The sampling positions and dimensions of the tensile specimens and microstructure samples are shown on the left side of Figure 3. Five groups of tensile specimens were located in four directions of the component, and the sampling directions were different. Tensile specimens A-1, A-2, B-2, and D were taken from the rib plate, and tensile specimen C was taken from the horizontal plane. The tensile test was carried out on an Instron-5569 (Instron, Boston, MA, USA) universal testing machine with an extensometer. The tensile speed was 1 mm/min. The optimum process parameters were determined by a range analysis of ultimate tensile strength and elongation, and the optimum process parameters were verified by experiments.

The microstructures were observed by an Olympus GX71 (Tokyo, Japan) optical microscope (OM). The microstructure samples were grounded with 400#, 800#, 1000#, and 1500# SiC papers and then polished by a diamond reagent with a particle size of 0.5 μm and a real velvet polishing cloth in a polishing machine until the mirror surface was smooth and free of scratches. After polishing, the samples were etched in Keller’s etchant (2.5% HNO_3_, 1.5% HCl, 1% HF, and 95% H_2_O) for 12 s. The size of a microstructure sample is shown in Figure 3. A Zeiss Merlin Compact scanning electron microscope (SEM, Zeiss, Jena, Germany) with energy dispersive spectroscopy (EDS, Zeiss, Jena, Germany) was used to analyze the fracture morphology and element distribution.

Meanwhile, the microstructures and properties at positions with local pressurization were emphatically analyzed. The positions were selected on side A and side B, which were named according to their relative positions for the convenience of narration. The sampling positions and dimensions of microstructure samples and tensile specimens at positions with local pressurization are shown on the right side of Figure 3.

## 3. Results and Discussion

### 3.1. Surface Quality and Internal Macro-Morphology

The surfaces of all formed components were observed, and the surface quality was good. The silver metallic luster and the streamline of the metal flow could be observed. The filling was complete, and no serious defects such as insufficient pouring or macroscopic cracks were found, as shown in the sample in Figure 4a–f. However, surface quality problems with misrun and peeling and cold lines at specific positions were observed, as shown in Figure 5a–c.

The problem of misrun was due to the insufficient filling capacity, which could be seen in the thin wall below position A-1, which was located at the far end of the sprue, so it was difficult to fill completely. The specific pressures of the No. 1, No. 2, No. 10, No. 11, and No. 13 components with the problem of misrun was less than 48 MPa, and the pouring temperature was 650 °C, so the filling capacity provided by the process parameters was insufficient. The peeling problem occurred in the deep cavity, corresponding to the raised structure of the mold. Due to the local heat accumulation, the local temperature of the mold was higher, so the melt on the surface failed to completely solidify, and the curvature of these positions was large, finally, the surface stress led to peeling after opening the mold. The cold lines were caused by the low temperature of the melt at the front and the incomplete fusion when the multi-strand melts met. The cold lines led to a reduction in the bonding strength, and the thin-wall parts were more prone to cold lines. Appropriately raising the mold temperature and pouring temperature and shortening the filling time can reduce the cold lines. These surface quality problems can be effectively solved by grinding and shot blasting in a later stage.

Shrinkage cavities and shrinkage porosity are common casting defects. The reason for this is that the volume difference caused by the isolated liquid phase solidification shrinkage cannot be supplemented by the surrounding liquid phase. Furthermore, shrinkage cavities and shrinkage porosity generally occur in thick-walled positions. This is because thick-walled positions easily form the thermal center, for instance, the final solidification area. In order to observe the internal quality of the components, six positions (A-1, A-2, A-3, B-1, B-2, and B-3) were dissected along the center of the deep hole on side A and side B, and the pressure delay times on side A and side B were different, as shown in Figure 6.

The macroscopic morphologies of positions A-1, A-3, B-1, and B-3 with and without local pressurization were compared. The area of the shrinkage cavity and shrinkage porosity were different in different zones, among which the area of A-1 was the largest. The shrinkage cavity and shrinkage porosity in the B-1 position were eliminated completely after local pressurization was applied. The shrinkage cavity and shrinkage porosity defects could be effectively eliminated, but not completely, by local pressurization in the experiments in this work. On the premise that the local specific pressure was sufficiently large, the pressure delay time was the key parameter for eliminating defects. Similarly, the defects in position A-3 still existed due to the short pressure delay time. The scope of forced feeding was limited, and the longer the distance was from the feeding point, the worse the feeding effect was. The defects at the lower left of position B-3 could not be eliminated, which corresponded to the area where the peeling problem occurred. The results of density measurements in the study by Hashemi et al. [18] stated that there was an achievable critical pressure. All squeeze casting specimens were completely dense at a pressure above 50 MPa. However, in this work, due to the equipment and safety factors, the maximum squeeze specific pressure was 48 MPa, the size of the flywheel housing component was large, and the process of pressure transmission was long, resulting in an obvious pressure loss at the far end. Although the local specific pressure was 800 MPa, the action area of the local pressurization rod was small, resulting in a limited action range. To sum up, local pressurization could lead to effective forced feeding, though it could not eliminate the shrinkage cavities and porosity defects in some positions. It was very beneficial to improve the mechanical properties of the local pressurization positions and the whole components.

### 3.2. Mechanical Properties and Fracture Morphology

Ultimate tensile strength and elongation are the key mechanical properties in this work. According to the histograms of mechanical properties of each component in Figure 7, it can be observed that the average ultimate tensile strength of the No. 3 component sample was 195 MPa, which was the best, and the data on the average ultimate tensile strength were mainly concentrated around 175 MPa; the worst was 165 MPa. The average yield strength was about 115 MPa, and the best was 121 MPa. The average elongation ranged from 1.5% to 3.5%. The average elongation of the No. 3 component sample was 3.13%, and the best was the No. 11 component sample. It was judged from the data that the corresponding process parameters of the No. 3 component were better.

In order to explore the differences in mechanical properties in different zones, the average and standard deviation values of ultimate tensile strength and elongation in five positions were calculated, and the values are listed in Table 4. By comparing the data, it was found that the zonal effect of the mechanical properties of the formed flywheel housing components was obvious. Among the five positions, the average ultimate tensile strength and the elongation at position C were about 180 MPa and 2.9%, which were better than those at the other positions. The mechanical properties of positions A-1 and A-2 were similar, while the mechanical properties of position B-2 were poor.

Clear cold shuts, pores, shrinkage porosity, and shrinkage cavities could be observed on the fractures of some tensile specimens, resulting in low strength and elongation or even invalid data. Figure 8 shows the SEM images of the fracture morphology of the tensile samples. Figure 8a–e show SEM images of the fractures of the 4-B-2, 5-D, and 16-C tensile samples; Figure 8a,c,e show the images at low magnification, and Figure 8b,d,f show the images at high magnification. Cleavage planes, tearing ridges, and a small number of dimples can be clearly observed in the figures.

The tensile fracture morphologies of the three groups of samples with different mechanical properties were compared. Multiple cold shuts, shrinkage cavities, and pores can be observed in Figure 8a. The existence of cold shuts led to the discontinuity of the grain boundary and the cleavage of the matrix bonding. Therefore, the interface bonding strength was low, and it was easily destroyed first under the action of force. The cleavage cracks occurred easily at the shrinkage cavities. The existence of cold shuts, shrinkage cavities, and pores reduced the mechanical properties [19]. Dense tearing ridges and dimples caused by plastic deformation can be observed in Figure 8b. The low average mechanical properties of position B-2 were confirmed by the observation and analysis of fracture morphology. In Figure 8c, several large shrinkage cavities and pores with a size of about 50 μm can be observed. The cleavage planes were concentrated and large, and the bright tearing ridges were not dense, so the strength and elongation of position D were at a low level. The research results of Abou El-Khair [20] also indicated that porosity had an adverse effect on the mechanical properties of aluminum alloy components, especially elongation. Compared with Figure 8c, the sizes of the cleavage planes shown in Figure 8d were smaller, the bright tearing ridges and dimples were denser, the layering was more obvious, and the performance of position C was better.

The fracture form was quasi-cleavage, which belongs to a range of brittle fractures and is a form of transition fracture between brittle fracture and ductile fracture. Quasi-cleavage fracture is a discontinuous fracture process in which many small cleavage cracks are generated in different parts and then grow under the action of force; the previous small cracks grow into cleavage planes and finally tear the residual parts of the connection in the form of plastic deformation, which is characterized by tearing ridges or dimples [21].

### 3.3. Orthogonal Experimental Analysis and Verification

The range analysis of the dual index for the ultimate tensile strength and elongation of the flywheel housing components was carried out (Table 5). I was the pouring temperature, II was the specific pressure, III was the mold temperature, IV was the local specific pressure, V was the pressure delay time, VI and VII were empty, $\sigma_{b}$ was the ultimate tensile strength, and δ was elongation.

#### Range Analysis

The average values of the data of each test index (i.e., the ultimate tensile strength and elongation) under each factor and level were calculated, and the range value (R) was calculated. The calculation results are shown in Table 6. According to the average values of different levels of ultimate tensile strength, the optimum process parameters were determined, which involved a pouring temperature of 650 °C, a specific pressure of 48 MPa, a mold temperature of 220 °C, and pressure delay times of 15 s (side A) and 17 s (side B). Similarly, according to the average value of different levels of elongation, the optimum process parameters were also given, which involved a pouring temperature of 650 °C, a mold temperature of 220 °C, a specific pressure of 40 MPa, and pressure delay times of 15 s (side A) and 17 s (side B).

Combined with the effect order of the factors, the optimum process parameters were determined by comprehensive consideration: a pouring temperature of 650 °C, a specific pressure of 48 MPa, a mold temperature of 220 °C, a local specific pressure of 800 MPa, and pressure delay times of 15 s (side A) and 17 s (side B). This corresponded to the No. 3 component. 

According to the optimum process parameters above, a repeated validation experiment was carried out. The stress–strain curves of tensile specimens of the validation experiment are shown in Figure 9. The mechanical properties at different positions from the validation experiment are shown in Table 7. The average ultimate tensile strength, yield strength, and elongation of the formed components according to the validation experiments on the optimum process parameters were 201 MPa, 103 MP, and 5.1%. The correctness of the optimum combination determined by the above analysis was verified by comparing the mechanical properties in Table 4 and Table 7.

### 3.4. Microstructure

The nucleation and growth of grains were different in the solidification process under different process parameters, which led to different microstructures. The microstructure samples were taken from the formed component with the optimum process parameters to analyze the microstructure. The sampling position of the microstructure samples is shown in Figure 3.

As shown in Figure 10a,b, the grains of the primary α-Al phase were mainly rosette and equiaxed grains; the grains were fine, and the average grain size was about 40 μm. Furthermore, except for some obvious cast microstructure (i.e., dendrites), some equiaxed grains and even spheroidal grains were also found in Figure 10. This was significantly different from the common microstructure consisting of dendrites. It was attributed to the effect of the applied pressure during squeeze casting.

AlSi9Mg aluminum alloy is a solidification shrinkage alloy. The critical nucleation radius ($r_{c}$) of the squeeze casting under thermodynamic conditions is as follows [22]:(1)$$r_{c}=\frac{2\sigma_{LS}T}{L\Delta T+K_{0}\varepsilon Tp}$$
where $r_{c}$ is the critical nucleation radius, $\sigma_{LS}$ is the solid–liquid interfacial tension, T is the solidification temperature, L is the latent heat of crystallization, ∆T is the supercooling, $K_{0}$ is a coefficient, ε is the volumetric shrinkage, and p is the specific pressure.

Due to the specific pressure (p) applied in squeeze casting, the critical nucleation radius decreased significantly, and more atomic groups participated in nucleation during solidification.

The kinetic conditions of solidification of aluminum alloy are the nucleation rate and the linear velocity of growth, in which the nucleation rate is an important factor affecting grain refinement. The nucleation rate refers to the number of crystal nuclei formed by liquid metal in the unit time and unit volume. The nucleation rate (N) is expressed as [23]:(2)$$N=\frac{nKT}{h}exp\left(-\frac{∆G}{KT}\right)exp\left(-\frac{∆G_{A}}{KT}\right)$$
where n is the number of atoms per unit volume, K is the Boltzmann constant, h is the Planck constant, ∆G is the nucleation energy, and $∆G_{A}$ is the diffusion activation energy. For the spherical nuclei, ∆G and $∆G_{A}$ are, respectively, as follows [24]:(3)$$∆G=\frac{16{\sigma_{LS}}^{3}T^{2}}{3{\left(L\Delta T+K_{0}\varepsilon Tp\right)}^{2}}$$
(4)$$∆G_{A}=∆G_{A0}\left(1+\beta p\right)$$
where $∆G_{A0}$ is the diffusion activation energy at atmospheric pressure and *β* is a coefficient.

The nucleation energy decreased and the atomic diffusion activation energy increased, so the nucleation rate increased significantly under the pressure of squeeze casting. Therefore, the grains were finer and more evenly distributed.

The grain growth mechanism at the rough interface was the continuous growth mechanism. The growth process was restricted by the surrounding grains, so the grains were fine. While some local grain sizes were larger, the dendrite arms were relatively developed. The morphology of silicon in the eutectic phase was acicular, the growth mode was cooperative growth mode, and the interior was interconnected and evenly distributed [25]. In addition, under the action of pressure, the silicon was prone to segregation, as shown in Figure 10c,d. In the segregation area, the silicon was finer in morphology, accompanied by fine dendrites and large block primary silicon particles. Abou El-Khair [20] also reported that the existence of acute eutectic silicon surrounded by α-Al dendrites was observed, as was shown in this work. They also indicated that the application of pressure did not affect the morphology of the eutectic silicon.

### 3.5. Microstructures and Properties at the Positions with Local Pressurization

#### 3.5.1. Mechanical Properties

According to the histograms of average mechanical properties at the different sampling positions in Figure 11, the average ultimate tensile strength values measured at different positions with local pressurization were greater than 180 MPa, and the values were concentrated around 200 MPa. The average yield strength values were concentrated around 110 MPa. The average ultimate tensile strength of the B-3-S sample with the best mechanical properties was 214 MPa, and the yield strength was 121 MPa. The strength values were relatively stable, and the fluctuation was not large. However, the numerical fluctuation of the elongation was obvious. The elongation of position A-3-S was 8.2%, while the elongation of position A-3-X was only 2.1%, with a high elongation difference of 6.1%. The fluctuation of the elongation was attributed to the existence of shrinkage cavities and pores. In addition, the morphology of the needle-like Si and α-Al grains also had an important influence on the mechanical properties of the formed parts. The needle-like Si split the α-Al matrix, resulting in a downward trend in elongation. The main reason was that the needle-like Si was a brittle phase. During the deformation process, the crack preferentially initiated in the needle-like Si, and the needle-like Si weakened the connection with the matrix. Spherical α-Al grains significantly improved the strength and elongation of the alloy, while dendritic α-Al grains led to a decrease in the mechanical properties of the alloy. There were differences in the microstructure of the flywheel shell at different positions, which was one of the reasons for the fluctuation of the strength and elongation of the alloy.

#### 3.5.2. Zonal Effect of Microstructure

According to the profile morphology after pressurization, the positions could be divided into a deformation zone, a direct action zone, and an indirect action zone. The deformation area was the area whose shape changed with the process of local pressurization. The division is shown in Figure 12.

The as-cast microstructure and plastic deformation microstructure coexisted in the deformation zone. The shear stress caused the plastic deformation, and the shear deformation in the process of local pressurization destroyed the continuity of the microstructure. The plastic deformation microstructure existed on the surface layer of the hole, and the plastic deformation in the deep layer became smaller and gradually transited to the as-cast microstructure.

Figure 13a,b show the as-cast microstructure in the deformation zone, and Figure 13c,d show the plastic deformation microstructure in the deformation zone. The as-cast microstructure was observed but not with coarse dendrites; rosette and equiaxed primary α-Al phase were mainly observed. The size of the fine grains was about 25 μm, and the maximum size was less than 100 μm. Hashemi et al. [18] also observed that dendrites solidified under squeeze casting pressure exhibited distinct zones of relative fine and coarse dendritic structure, which was similar to what was observed in this work. The secondary dendrite arm spacing (SDAS) and solidification time have a strict correlation, satisfying the following functional relationship [26]:(5)$$\lambda_{2}=c{\;\cdot \;T}_{f}^{1/3}$$
where c is the constant related to the alloy, $T_{f}$ is the local solidification time, Tf=ΔTRG, where R is the dendrite growth rate, and G is the temperature gradient. 

The temperature gradient of the surface layer was large, so the local solidification time was short, and the grain could not grow after the nucleation of α-Al phase. The morphology of the eutectic phase varied with the solidification time. The eutectic structure shown in Figure 13a was fine, while the eutectic silicon shown in Figure 13b was slender, needle-shaped, and mixed with some bulk silicon. An obvious deformation characteristic was observed at positions 3 and 4, as shown in Figure 13c,d.

As shown in Figure 14, the plastic deformation microstructures existed in both the direct action zone and the indirect action zone. In the direct action zone, it was generated by compressive stress. Larger compressive stress led to obvious plastic deformation in the I and II positions, as shown in Figure 14a,b. The grains were compressed into elongated shapes under the action of local pressure. Pressure loss occurred in the transfer process in the liquid–solid slurry. With an increase in the distance from the free surface, the plastic deformation decreased and the density decreased, and it finally transitioned to the as-cast microstructure in the IV position, as shown in Figure 14d. As the alloy was a continuous medium, the plastic deformation microstructure also existed in the indirect action zone, forming plastic deformation layers of different depths.

Characteristic crystal-phase microstructures existed at III position in the indirect action zone, as shown in Figure 14c. The grains of the α-Al phase were connected, the grain boundaries were not very clear, the number of grain boundaries was low, and the area of the eutectic phase was small and separated. The phase with a low melting point was forced to flow through the dendrite channel constructed by the dendrite arm under pressure. The scouring effect caused by strong flow fused the dendrite arm and became the new core of nucleation, which greatly improved the nucleation rate. Under the influence of undercooling, the grain growth rate was large, so the α-Al grains were connected into planes by points, and the area of the divided eutectic phase was small.

Voids produced by solidification shrinkage without the timely feeding of the liquid phase led to the generation of shrinkage cavities and shrinkage porosity. Shrinkage defects generally appeared in the final solidification area (i.e., hot spots), and the solidification mode was changed to paste solidification. As shown in Figure 15a, the final area of paste solidification had obvious boundaries, and the morphology of the α-Al phase was obviously different. The grains of the α-Al phase on the left had higher roundness and were mainly equiaxed and rosette grains, while the grains on the right were dendrite grains. Figure 15b shows the microstructure around a shrinkage cavity with developed dendrites, a large number of eutectic phases, and a high degree of segregation, mixed with lots of massive silicon particles. Thus, it can be seen that the solidification time at the defect locations was long, the eutectic phase with a low melting point was mostly concentrated here, and the silicon element was enriched.

#### 3.5.3. Analysis of Pressurization Effect

The effect of local pressurization is mainly reflected in the following three aspects: changing the solidification process, plastic deformation, and forced feeding.

(1)Changing the solidification process:

In most casting processes, an air gap is formed between the mold and the solidified shell of the casting shortly after pouring. It is caused by the simultaneous shrinkage of the shell and the expansion of the mold. The formation of an air gap changes the heat conduction of the heat transfer mechanism into convection and radiation, significantly decreasing the heat transfer rate and thus decreasing the cooling rate [27]. In squeeze casting and local pressurization, the pressure applied eliminates the formation of the air gap, which greatly improves the heat transfer and cooling rate [25].

In the process of local pressurization, due to the rapid cooling effect and pressure effect of the local pressurization rods, the solidification conditions of the casting, i.e., the temperature gradient (G) and solidification rate (v), were changed. In the solidification process, due to the enrichment or dilution of solute in front of the interface, the liquid phase equilibrium solidification temperature was reduced accordingly, resulting in constitutional supercooling. The criterion for constitutional supercooling is as follows [28]:(6)$$\frac{G}{v}<\frac{m_{L}C_{0}(1-k_{0})}{D_{L}k_{0}}$$
where G is the temperature gradient, v is the solidification rate, $C_{0}$ is the concentration of the primitive component, $k_{0}$ is the equilibrium distribution coefficient of solute concentration, $D_{L}$ is the diffusion coefficient of solute in the liquid phase, and $m_{L}$ is the liquidus slope.

Constitutional supercooling is the main factor determining the microstructure of the alloy. With a decrease in Gv, the crystal morphology changes from planar to dendrite [27], and the intervention of pressure generally promotes the formation of constitutional supercooling. The key factor affecting the growth morphology of a solid solution is the melting temperature gradient. With the progress of pressurization, the temperature gradient first increased and then decreased [29]. In the range of constitutional supercooling, a higher temperature gradient led to smaller constitutional supercooling, and a lower temperature gradient led to larger constitutional supercooling and a wide undercooling zone.

(2)Plastic deformation:

The microstructures at position A-3 with and without local pressurization were compared. As shown in Figure 16a, the grains were elongated, and obvious plastic deformation microstructure appeared, so the local properties were strengthened. The microstructure in Figure 16b was mostly composed of fine and uniform equiaxed grains, and there were few signs of columnar grains. The results agreed quite well with the results reported by Maleki et al. [30].

Map scanning of samples at position A-3 of the No. 3 and No. 6 experimental components was carried out to determine the distribution of the alloying elements of the squeeze casting components. Figure 17a shows the element distribution of the plastic deformation microstructure of the No. 3 components. Figure 17b shows the element distribution of the as-cast microstructure of the No. 6 components. According to the national standard, the mass fraction of the main elements in AlSi9Mg was within a reasonable range except that silicon was enriched to a certain extent. Al was the main matrix element, and Mg and Si were strengthening elements. Some silicon mainly existed in the form of the Al-Si eutectic phase. The other silicon could form Mg_2_Si strengthening phase, and Mn was used to reduce the Fe impurity that was introduced in the smelting process. It is important to accurately identify coarse iron-rich intermetallic compounds commonly found in Al-Si cast alloys, as some of these phases are associated with the deterioration of mechanical properties [31]. As shown in Figure 17, the grain size of No. 3 with local pressurization was obviously smaller than that of No. 6 without local pressurization. The Si element was mainly distributed at the grain boundaries, but the Al element at grain boundaries exhibited a smaller amount compared with that inside the grain. This was attributed to the formation of the Al-Si eutectic phase. The α-Al nucleated and grew normally, and the eutectic silicon was distributed around the grains, while there was more eutectic silicon in the final paste solidification zone. The distribution of Si was more chaotic after plastic deformation. Mg was observed to have a dispersion distribution without large-scale segregation. Mn and Fe existed at grain boundaries. Furthermore, these two elements always coexisted. This might have been due to the formation of Fe and Mn intermetallic.

The content of elements at different points was analyzed (shown in Table 8). The results showed that the contents of Fe and Mn were relatively high in the polygonal and brighter phases (i.e., point 1 and point 2), which was close to 1:1. The atomic numbers of Fe and Mn were quite different from those of matrix aluminum, so the Fe and Mn seen were relatively bright, while the atomic numbers of aluminum and silicon were close, and the contrast was not very obvious. The coexistence of Fe and Mn in the same phase and the absence of an acicular-shaped phase confirmed that Mn can neutralize the harmful effects of Fe. Obvious corrosion pits were observed at some grain boundaries after corrosion, indicating higher energy at grain boundaries. This illustrated that point 3 was inside the α-Al grain and the position of point 4 was at the grain boundary, according to the elemental composition analysis.

**Table 8 materials-17-00106-t008:** Energy spectrum analysis of fixed points in Figure 18 (wt %).

Points	Al	Si	Mg	Mn	Fe
1	60.90	11.03	0.00	14.74	13.34
2	63.47	9.19	0.03	13.38	13.93
3	97.28	2.31	0.16	0.25	0.00
4	59.56	40.17	0.12	0.05	0.11

**Figure 18 materials-17-00106-f018:**
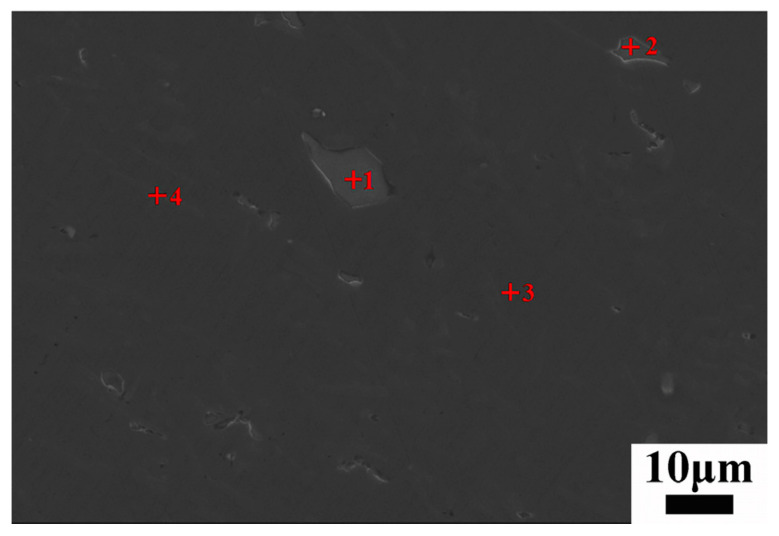
Scanning picture of position A-3 of the No. 3 component sample.

(3)Forced feeding:

The main purpose of local pressurization was to achieve forced feeding. During solidification, dendrites grow and connect into skeletons, hindering the free flow and feeding of liquid and resulting in shrinkage porosity and shrinkage cavities. The intervention of an external stress field forces the flow of the un-solidified liquid phase for forced feeding. The criterion formula for predicting shrinkage porosity on the basis of the Niyama criterion [32] is as follows:(7)$$\frac{G}{\sqrt{v}}=\sqrt{\frac{\eta .f_{L}.\beta }{k_{0}}.\frac{\Delta T}{\Delta P-m\Delta T.g/A}}\leq K$$
where η is the viscosity coefficient of the liquid phase, $f_{L}$ is the liquid fraction, β is the solidification shrinkage, ΔT is the temperature difference of the solidus-liquidus temperatures, g is the gravitational acceleration, A is the correction coefficient, and Δp is the pressure difference.

According to the metallographic structure in Figure 19, the eutectic phase with a low melting point was forced to flow by an external force; namely, the band-shaped segregation of the eutectic phase was observed between the deformed structure and the as-cast structure. The eutectic phase with a low melting point flowed through the dendrite channel and solidified after agglomeration.

## 4. Conclusions

The squeeze casting process and quality of AlSi9Mg aluminum alloy flywheel housing components with a large wall thickness difference and a complex shape were comprehensively evaluated as a whole. The microstructures and mechanical properties of the formed components were investigated, and the positions with local pressurization were emphatically analyzed. Some important conclusions are summarized as follows:The surface quality of the formed components was good. The effect of local pressurization for forced feeding on the elimination of the shrinkage cavity and shrinkage porosity was obvious, and the scope and effect of feeding were limited. The longer the distance was from the feeding point, the worse the effect of feeding was.Through the analysis, it was determined that the optimum combination of process parameters was a pouring temperature of 650 °C, a specific pressure of 48 MPa, a mold temperature of 220 °C, a local pressurization of 800 MPa, and pressure delay times of 15 s (side A) and 17 s (side B). The ultimate tensile strength, yield strength, and elongation of the formed components according to the validation experiment of the optimum process parameters were 201 MPa, 103 MPa, and 5.1%.The grains of primary α-Al were mainly rosette and equiaxed grains; the grains were fine, and the average grain size was about 40 μm. The microstructure of eutectic silicon was acicular, and silicon was prone to segregation under the action of pressure.According to the profile morphology, the positions after local pressurization could be divided into a deformation zone, a direct action zone, and an indirect action zone. The microstructures corresponding to different zones were different. The coexistence of as-cast and plastic deformation microstructures was observed. The effect of local pressurization on microstructures was mainly reflected in changing the solidification process, plastic deformation, and forced feeding. The local quenching caused by pressurization caused constitutional supercooling and then changed the morphology of the grains. Local pressurization caused local plastic deformation, and the grains were elongated. The strengthening element Mg was dispersed, and after plastic deformation, the distribution of Si was disordered. Local forced feeding was achieved by introducing large pressure differences through local pressurization.

## Figures and Tables

**Figure 1 materials-17-00106-f001:**
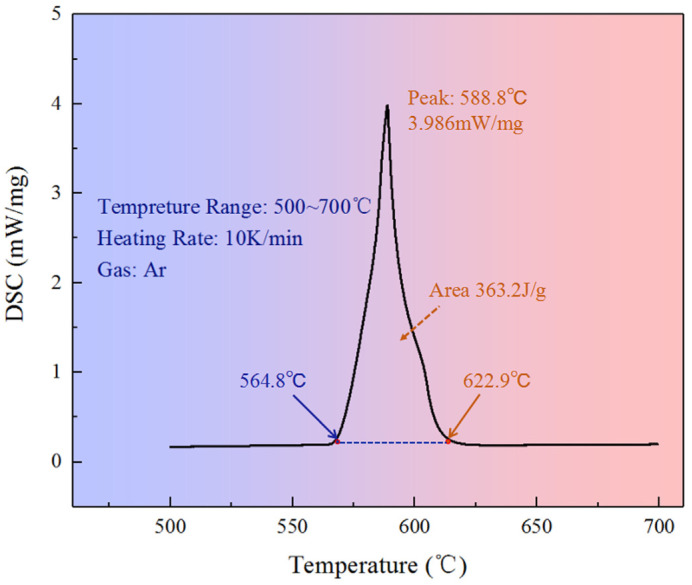
DSC curve of the AlSi9Mg aluminum alloy.

**Figure 2 materials-17-00106-f002:**
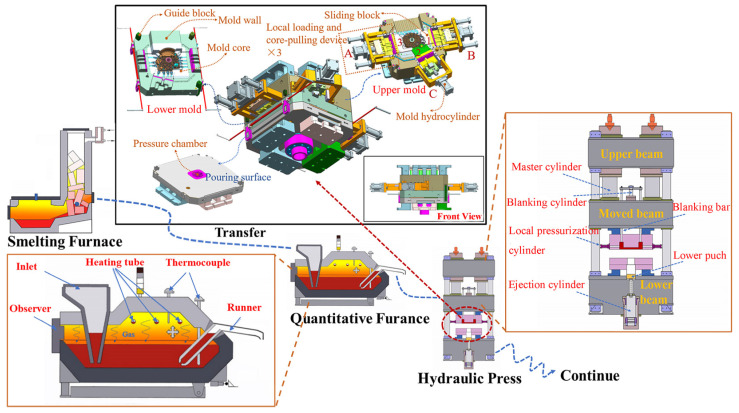
Squeeze casting flywheel housing process and explosion diagram.

**Figure 3 materials-17-00106-f003:**
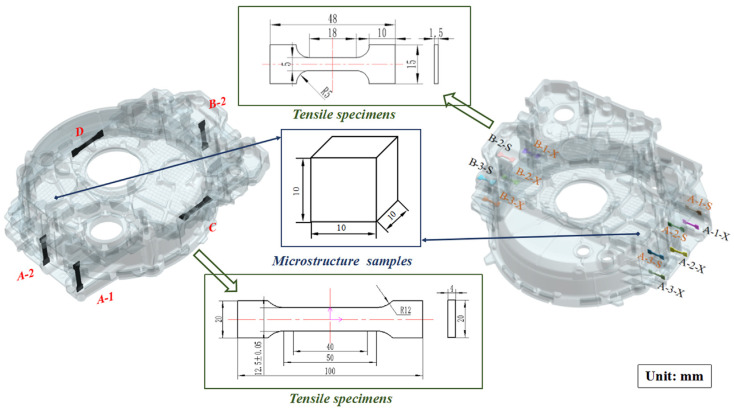
The sampling positions and dimensions of tensile specimens and microstructure samples.

**Figure 4 materials-17-00106-f004:**
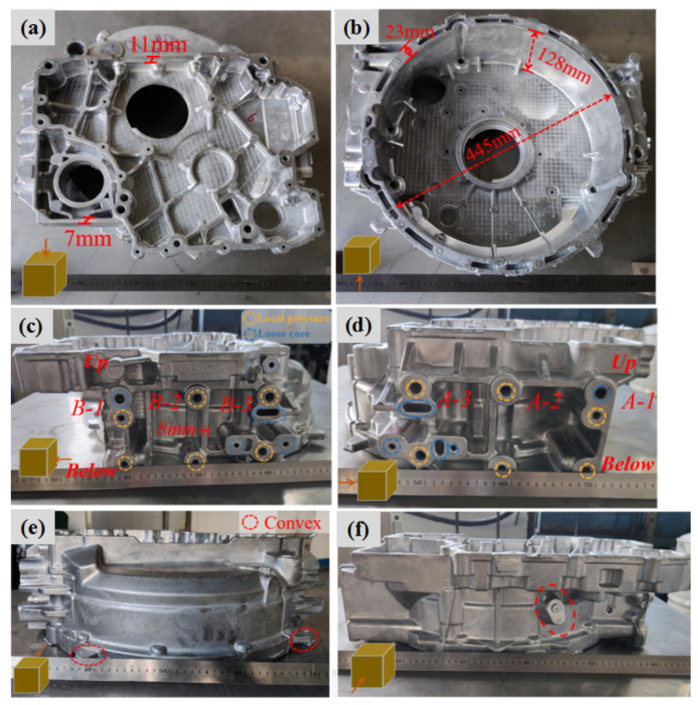
Appearance of a squeeze cast flywheel housing component: (**a**) top view, (**b**) bottom view, (**c**) right view, (**d**) left view, (**e**) front view, and (**f**) back view.

**Figure 5 materials-17-00106-f005:**
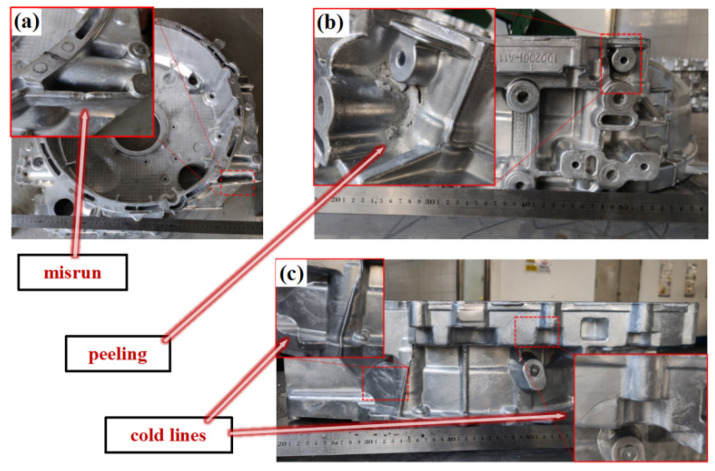
Surface quality problems: (**a**) misrun, (**b**) peeling, and (**c**) cold lines.

**Figure 6 materials-17-00106-f006:**
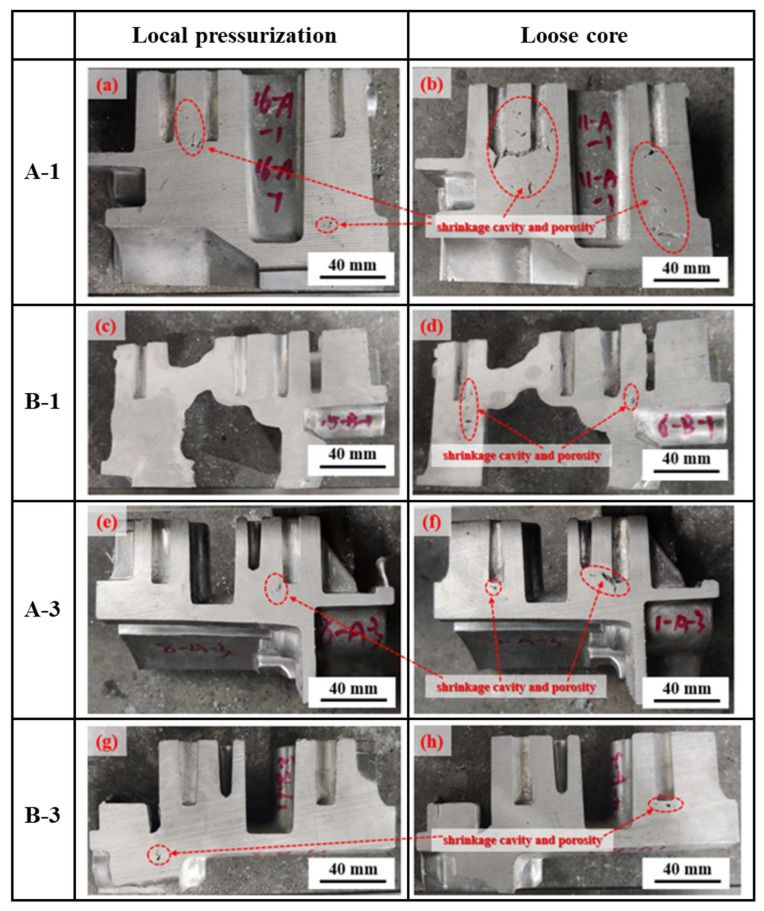
The profile macroscopic morphology of different positions of the formed components: (**a**,**c**,**e**,**g**) with local pressurization and (**b**,**d**,**f**,**h**) without local pressurization.

**Figure 7 materials-17-00106-f007:**
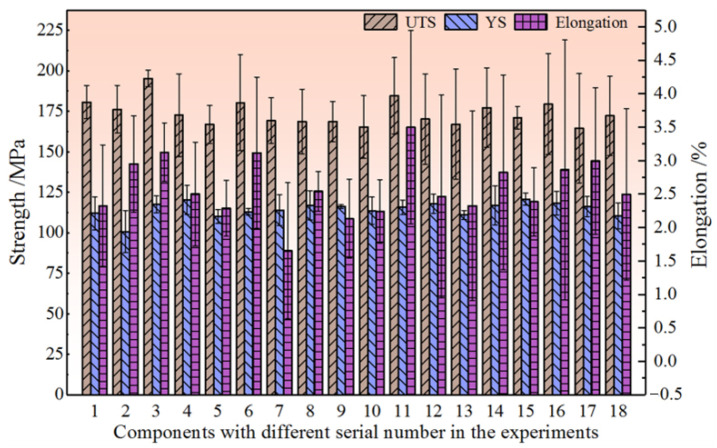
Histogram of average mechanical properties of each component.

**Figure 8 materials-17-00106-f008:**
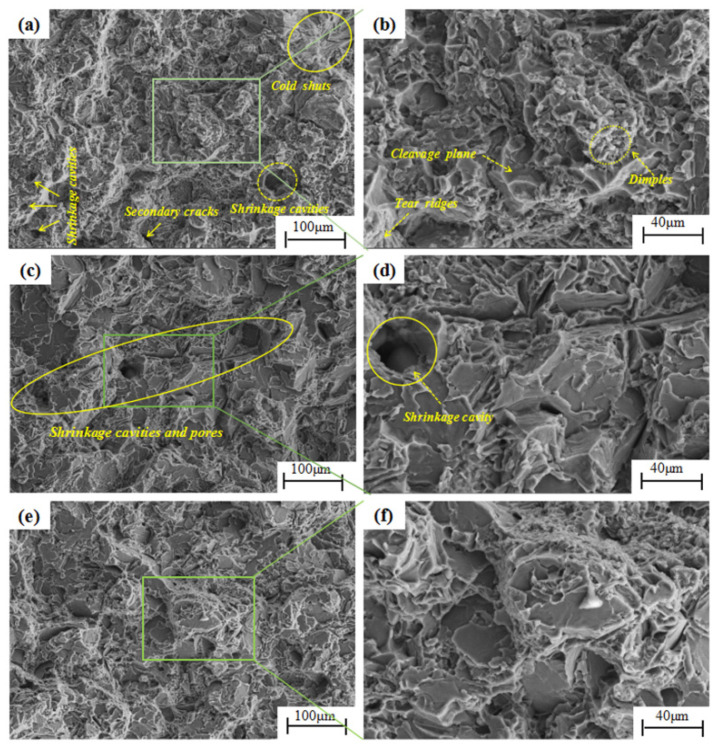
Comparison of fracture morphologies of (**a**,**b**) 4-B-2, (**c**,**d**) 5-D, and (**e**,**f**) 16-C tensile specimens with different mechanical properties.

**Figure 9 materials-17-00106-f009:**
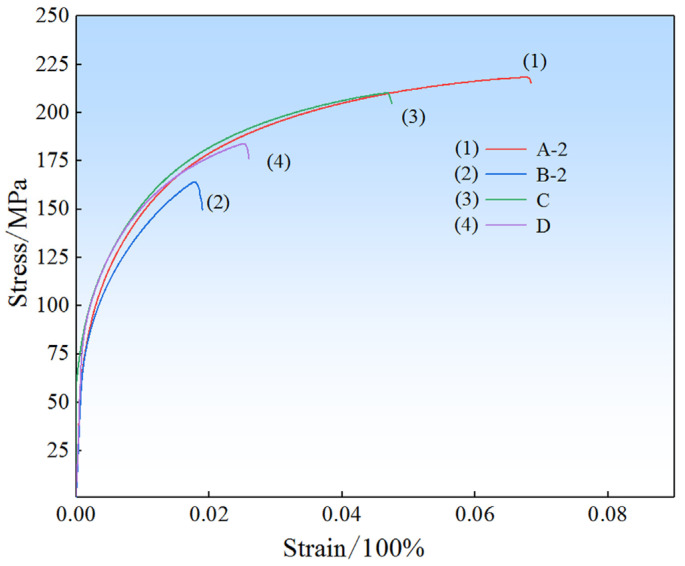
Stress–strain curves of tensile specimens in the validation experiment.

**Figure 10 materials-17-00106-f010:**
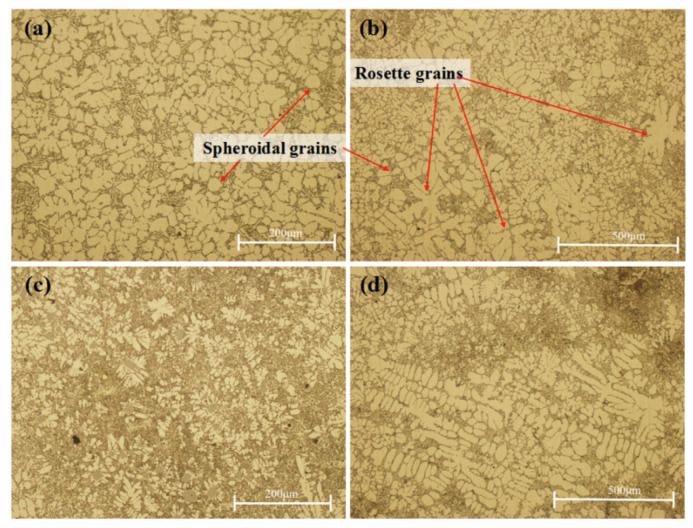
Microstructures at the sampling positions of the formed component with the optimum process parameters: (**a**,**b**) are microstructures at 100× and 200× magnification; (**c**,**d**) are microstructures of different types of silicon segregation.

**Figure 11 materials-17-00106-f011:**
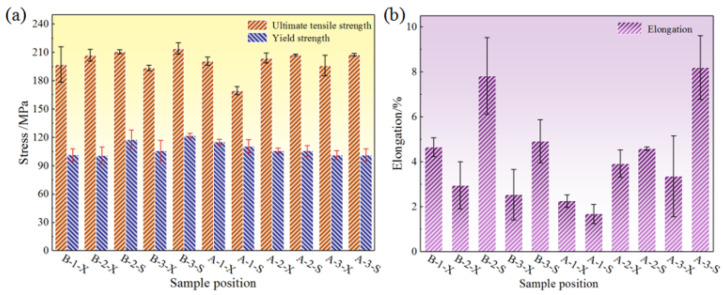
Histograms of average mechanical properties at different positions with local pressurization: (**a**) strength and (**b**) elongation.

**Figure 12 materials-17-00106-f012:**
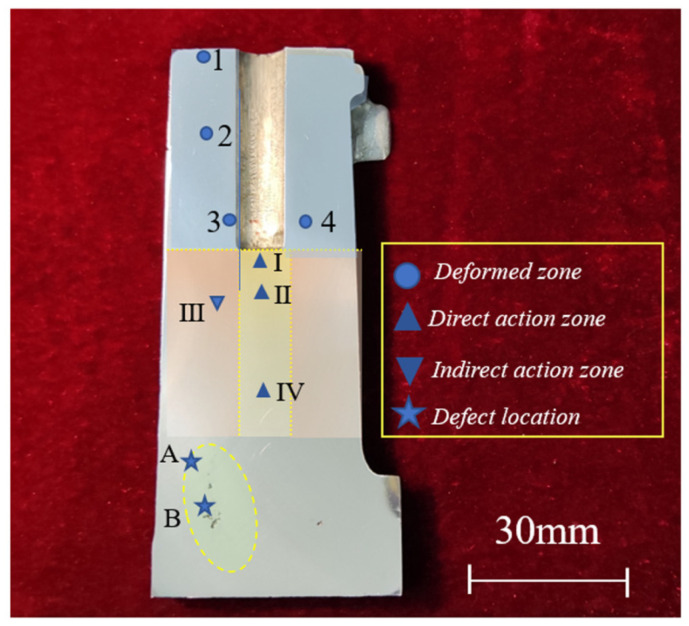
Zone division and sampling positions.

**Figure 13 materials-17-00106-f013:**
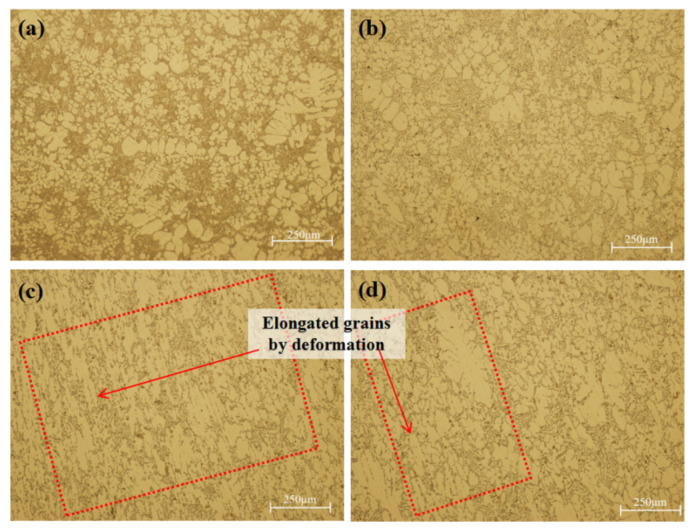
Microstructure of sampling positions (**a**) 1, (**b**) 2, (**c**) 3, and (**d**) 4 in the deformation zone.

**Figure 14 materials-17-00106-f014:**
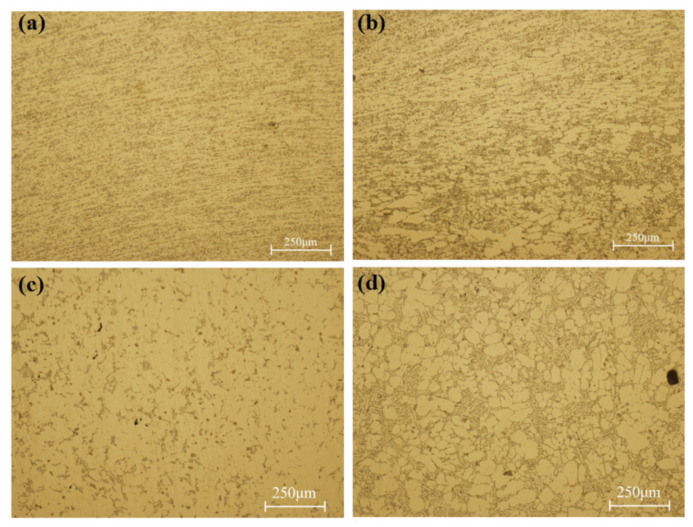
Microstructures of sampling positions (**a**) I, (**b**) II, and (**d**) IV in the direct action zone and (**c**) III in the indirect action zone.

**Figure 15 materials-17-00106-f015:**
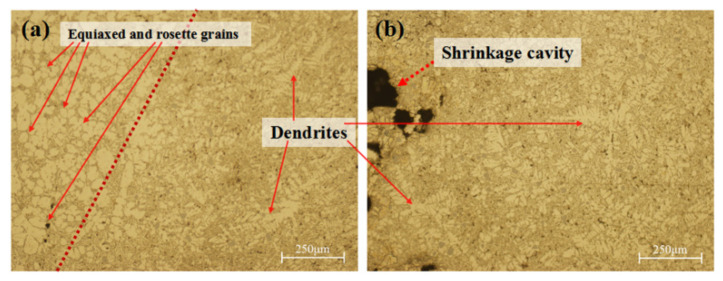
Typical microstructures of defect locations from sampling positions (**a**) A and (**b**) B in the zone division in Figure 12.

**Figure 16 materials-17-00106-f016:**
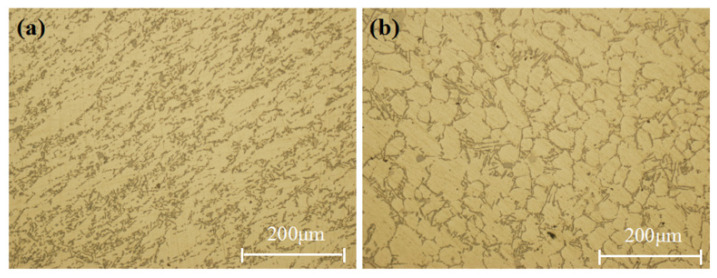
Microstructure of sampling position A-3 from (**a**) the No. 3 component with local pressurization and (**b**) the No. 6 component without local pressurization.

**Figure 17 materials-17-00106-f017:**
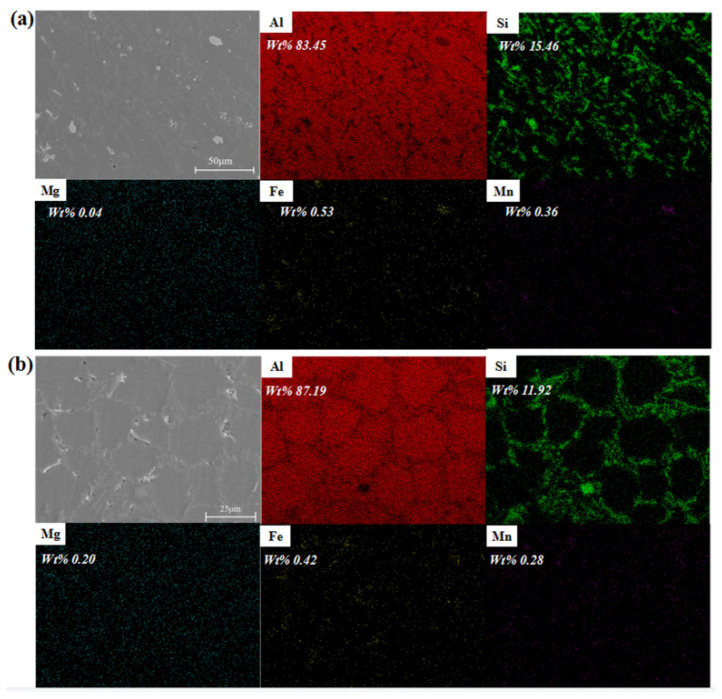
Element distribution at sampling position A-3 of (**a**) the No. 3 component with local pressurization and (**b**) the No. 6 component without local pressurization.

**Figure 19 materials-17-00106-f019:**
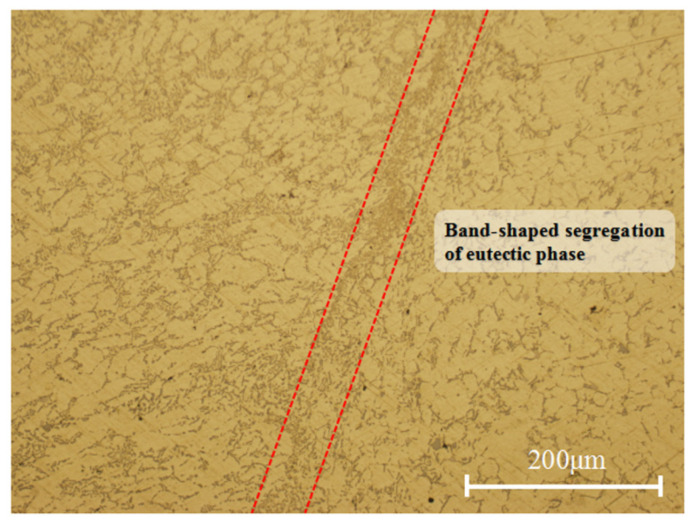
Metallographic structure under forced feeding.

**Table 1 materials-17-00106-t001:** Chemical composition of AlSi9Mg aluminum alloy (wt %).

Alloy	Si	Mg	Mn	Cr	Al	Fe	Cu	Zn
AlSi9Mg	8.79	0.194	0.246	0.03	Bal.	0.295	0.074	0.027

**Table 2 materials-17-00106-t002:** Five factors and three levels during the squeeze casting process of aluminum alloy flywheel housing components.

	Pouring Temperature/°C	Specific Pressure/MPa	Mold Temperature/°C	Local Specific Pressure/MPa	Pressure Delay Time/s
1	650	32	190	0	10 (side A); 12 (side B)
2	658	40	210	800	13 (side A); 15 (side B)
3	665	48	220	800	15 (side A); 17 (side B)

**Table 3 materials-17-00106-t003:** Orthogonal experiment scheme during the squeeze casting process of aluminum alloy flywheel housing components.

Serial Numbers	Pouring Temperature/°C	Specific Pressure/MPa	Mold Temperature/°C	Local Specific Pressure/MPa	Pressure Delay Time/s
1	650	32	190	0	10 (side A); 12 (side B)
2	650	40	210	800	13 (side A); 15 (side B)
3	650	48	220	800	15 (side A); 17 (side B)
4	658	32	190	800	13 (side A); 15 (side B)
5	658	40	210	800	15 (side A); 17 (side B)
6	658	48	220	0	10 (side A); 12 (side B)
7	665	32	210	0	15 (side A); 17 (side B)
8	665	40	220	800	10 (side A); 12 (side B)
9	665	48	190	800	13 (side A); 15 (side B)
10	650	32	220	800	13 (side A); 15 (side B)
11	650	40	190	0	15 (side A); 17 (side B)
12	650	48	210	800	10 (side A); 12 (side B)
13	658	32	210	800	10 (side A); 12 (side B)
14	658	40	220	0	13 (side A); 15 (side B)
15	658	48	190	800	15 (side A); 17 (side B)
16	665	32	220	800	15 (side A); 17 (side B)
17	665	40	190	800	10 (side A); 12 (side B)
18	665	48	210	0	13 (side A); 15 (side B)

**Table 4 materials-17-00106-t004:** Average values of mechanical properties at different positions.

	A-1	A-2	B-2	C	D
Ultimate tensile strength/MPa	177.4 ± 26.8	175.0 ± 14.7	167.3 ± 13.7	179.5 ± 23.7	168.7 ± 17.4
Elongation/%	2.65 ± 1.21	2.58 ± 1.03	1.56 ± 0.81	2.92 ± 1.32	2.39 ± 1.15

**Table 5 materials-17-00106-t005:** Results of orthogonal experiments of aluminum alloy flywheel housing components formed by squeeze casting.

	I/°C	II/MPa	III/°C	IV/MPa	V/s	*σ_b_*/MPa	*δ*/%
1	650	32	190	0	A	180.7	2.33
2	650	40	210	800	B	176.3	2.95
3	650	48	220	800	C	195.3	3.13
4	658	32	190	800	B	172.7	2.50
5	658	40	210	800	C	167.0	2.29
6	658	48	220	0	A	180.3	3.12
7	665	32	210	0	C	169.3	1.65
8	665	40	220	800	A	168.7	2.54
9	665	48	190	800	B	168.7	2.13
10	650	32	220	800	B	165.3	2.24
11	650	40	190	0	C	184.7	3.50
12	650	48	210	800	A	170.3	2.46
13	658	32	210	800	A	167.0	2.33
14	658	40	220	0	B	177.3	2.83
15	658	48	190	800	C	171.0	2.39
16	665	32	220	800	C	179.5	2.86
17	665	40	190	800	A	164.5	2.99
18	665	48	210	0	B	172.2	2.50

**Table 6 materials-17-00106-t006:** Range analysis of orthogonal experiment of aluminum alloy flywheel housing components formed by squeeze casting.

		I	II	III	IV	V
Ultimate tensile strength/MPa	K1	178.77	172.42	173.72	177.42	171.92
	K2	172.55	173.08	170.35	173.08	172.08
	K3	170.48	176.30	177.73	171.30	177.80
	R	8.29	3.88	7.38	6.12	5.88
Elongation/%	K1	2.77	2.32	2.64	2.66	2.63
	K2	2.58	2.85	2.36	2.62	2.53
	K3	2.45	2.62	2.79	2.52	2.64
	R	0.32	0.53	0.43	0.14	0.11

**Table 7 materials-17-00106-t007:** Mechanical properties at different positions in the validation experiment.

	A-1	A-2	B-2	C	D	Average
Ultimate tensile strength/MPa	230.6	218.4	164.2	210.2	183.9	201.5
Yield strength/MPa	104.8	102.8	99.4	90.5	112.9	102.1
Elongation/%	7.15	7.44	2.17	5.58	3.23	5.11

## Data Availability

The data that support the findings of this study are available from the corresponding author upon reasonable request.

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
