# Peer review of "Effect of Local Pressurization on Microstructure and Mechanical Properties of Aluminum Alloy Flywheel Housing with Complex Shape"

_materials, 2023, doi:10.3390/ma17010106_

Round 1
Reviewer 1 Report
Comments and Suggestions for Authors
This study investigates the squeeze casting of AlSi9Mg aluminum alloy flywheel housings with complex shapes and varying wall thicknesses, focusing on how different process parameters affect defects, microstructures, and mechanical properties. It is seen that the study has been prepared in a detailed and devoted manner. It was found sufficient in terms of language.
However, there are a few aspects that I believe need further attention and clarification within your study. Please find them listed below:
(1) How did authors select the specific process parameters (e.g., pouring temperature, specific pressure, mold temperature) for your experiments?
(2) Why was AlSi9Mg aluminum alloy specifically chosen for these experiments?
(3) How do the defects observed in this study compare with those typically found in similar casting processes?
(4) It is recommended that authors elaborate on how the microstructures, especially the acicular silicon and primary 𝛼-Al grains, impact the mechanical properties of the cast components.
(5) it's indeed beneficial to cite recent studies to demonstrate the current state of knowledge in the field. Recent studies provide the most up-to-date findings and theories, which can help in positioning your research within the context of ongoing academic discourse..
(6) How do authors envision the application of the results of the study in an industrial setting?
(7) Are there any challenges or considerations regarding the scalability of this process for mass production?
(8) present certain figures (Figures 3, 6, 11) in a larger size is constructive and significantly enhance the readability and comprehension of the material.
In addition, if it is available, include a pattern for Figure 18.
Author Response
Response to Reviewers
No. materials-2742909
Title: Microstructures and mechanical properties of aluminum alloy flywheel housing components with large wall thickness difference and complex shape formed by squeeze casting with local pressurization
Dear editors and reviewers,
Thank you for your approval and valuable comments on our manuscript. These comments are all valuable and helpful for revising and improving our paper. We have studied these comments carefully and have made necessary revision. We hope the current status of the revised manuscript can meet the requirements. The revised parts are marked in yellow in the “Revised Manuscript”. The specific responses to the comments are list below:
The responses to the reviewers’ comments are listed below:
Reviewer #1: This study investigates the squeeze casting of AlSi9Mg aluminum alloy flywheel housings with complex shapes and varying wall thicknesses, focusing on how different process parameters affect defects, microstructures, and mechanical properties. It is seen that the study has been prepared in a detailed and devoted manner. It was found sufficient in terms of language.
However, there are a few aspects that I believe need further attention and clarification within your study. Please find them listed below:
(1) How did authors select the specific process parameters (e.g., pouring temperature, specific pressure, mold temperature) for your experiments?
Response: We have employed numerical simulation method to select process parameters. The numerical simulation in the early stage of forming can well show us the forming state under different process parameters and predict the possible location of defects. In order to obtain a well-formed flywheel shell part, we have done a lot of numerical simulation research in the early stage of the experiment. Finally, the best experimental process scheme was determined by numerical simulation. The previous numerical simulation research has also been published {https://doi.org/10.3390/ma15124334}.
(2) Why was AlSi9Mg aluminum alloy specifically chosen for these experiments?
Response: As an important representative of near-eutectic casting aluminum alloy, AlSi9Mg casting aluminum alloy has excellent casting properties and mechanical properties. It has been widely used in aerospace equipment, aviation equipment, mechanical engineering and automotive engineering. This shows that AlSi9Mg aluminum alloy has a more mature technology in engineering application. In addition, squeeze casting forming requires the alloy to have good fluidity to ensure that the alloy melt has good fluidity during the forming process. The addition of Si can significantly improve the fluidity of the alloy. Excessive Si content can lead to oxidation of the alloy at high temperatures. Squeeze casting, also known as liquid forging, allows solidified alloys to undergo plastic deformation under pressure. And lower Si content is more conducive to plastic deformation. Studies have shown that the addition of Mg can increase the elongation and tensile strength of aluminum alloys. The addition of Mn, Cu and Fe can significantly improve the oxidation resistance of the alloy. According to Fig.1, AlSi9Mg alloy has a wide solid-liquid region, which is very favorable for the filling of the workpiece. Based on the above factors, AlSi9Mg alloy is a reasonable choice.
(3) How do the defects observed in this study compare with those typically found in similar casting processes?
Response: Fig. 6 shows macroscopic local sectional photos of local pressure squeeze casting and traditional squeeze casting. Fig. 6 (a), (c), (e) and (g) show the macroscopic morphology of the local pressure flywheel housing. Fig. 6(b), (d), (f) and (h) show the local cutting macroscopic morphology of the typical squeeze casting flywheel housing. By comparison, it can be found that local pressure can significantly inhibit the formation of defects, and a large number of large-sized shrinkage defects appear in different parts of the typical squeeze casting flywheel housing.
(4) It is recommended that authors elaborate on how the microstructures, especially the acicular silicon and primary ?-Al grains, impact the mechanical properties of the cast components.
Response: In this paper, we add the discussion on the effect of needle-like Si and α-Al grains on the mechanical properties. The needle-like Si will reduce the mechanical properties of the alloy as the crack initiation position. The morphology of α-Al grains has an important influence on the mechanical properties. Excessive dendritic α-Al grains cause a decrease in mechanical properties. We also explain the reason why the mechanical properties of the flywheel housing fluctuate at different positions.
(5) It's indeed beneficial to cite recent studies to demonstrate the current state of knowledge in the field. Recent studies provide the most up-to-date findings and theories, which can help in positioning your research within the context of ongoing academic discourse.
Response: We cited the latest research on large aluminum alloy flywheel housing in the introduction. The latest research explored the effect of T6 heat treatment on the microstructure and mechanical properties of ZL204 alloy flywheel housing. This study mainly introduced the influence of different heat treatment parameters on the mechanical properties of alloys, and obtained the optimal heat treatment process parameters and mechanical properties. The influence of heat treatment process on eutectic Si was introduced, and the recrystallization mechanism of ZL204 alloy flywheel housing was revealed through EBSD.
(6) How do authors envision the application of the results of the study in an industrial setting?
Response: At the beginning of the design, the flywheel housing was designed according to the requirements of industrial applications. We designed three flywheel housings with different weights and sizes according to the requirements of industrial applications. The flywheel housing shown in this study was only one of the three flywheel housings. After the completion of the product and mold design, we first conducted extensive simulations to ensure the occurrence of defects in the actual production process. At the same time, we carried out a large number of forming experiments under different conditions and obtained a large number of experimental samples. Through a large number of sample samples, we obtained the best process parameters for actual production. At present, the flywheel housing shown in this study has established a production line at Dalian Innovation Die Casting Co., Ltd. and achieved mass production.
(7) Are there any challenges or considerations regarding the scalability of this process for mass production?
Response: There are two main challenges in this study: one is that there are many limitations on large-scale squeeze casting equipment, and the existing squeeze casting equipment is difficult to achieve the required tonnage. The lack of equipment is very difficult to achieve large-size flywheel shell forming. On the other hand, our formed flywheel housing has complex shape, large size and large wall thickness difference, which poses a greater challenge to the forming of flywheel housing. In view of the above two problems, we first designed a 12500 kN large-scale squeeze casting equipment to break the technical monopoly, and then proposed a local loading local feeding technology to successfully solve the defect problem of the formed parts. In addition, the 12500 kN large-scale squeeze casting equipment significantly reduces the equipment cost compared to other large-scale squeeze casting equipment. This study has done a lot of research work on the second problem and challenge.
(8) Presenting certain figures (Figures 3, 6, 11) in a larger size is constructive and significantly enhance the readability and comprehension of the material. In addition, if it is available, include a pattern for Figure 18.
Response: Fig. 3, 6 and 11 are enlarged in size and the width of the image is enlarged from 9 cm to 14 cm. We have adjusted the picture pattern of Fig. 18. The microstructure in Fig. 18 is clearly shown.
We hope that the revised manuscript can meet the requirements of the reviewers and the journal. We are looking forward to hearing your response soon. Thanks again.
Yours sincerely,
Dr. and Prof. Jufu Jiang

Reviewer 2 Report
Comments and Suggestions for Authors
A successful work, with only minor criticisms that do not detract from the very good quality of the work. Due to the many bright colors, for example in Fig. 2, some texts may be difficult to read in the illustrations.
Reviewer 3 Report
Comments and Suggestions for Authors
I have provided my comments on the quality of English in the review report.
